# Rare genomic copy number variants implicate new candidate genes for bicuspid aortic valve

**Steven G. Carlisle**[1], **Hasan Albasha**[2], **Hector I. Michelena**[3], **Anna Sabate-Rotes**[4], **Lisa Bianco**[4], **Julie De Backer**[5], **Laura Muiño Mosquera**[5], **Anji T. Yetman**[6], **Malenka M. Bissell**[7], **Maria Grazia Andreassi**[8], **Ilenia Foffa**[8], **Dawn S. Hui**[9], **Anthony Caffarelli**[10], **Yuli Y. Kim**[11], **Dongchuan Guo**[1], **Rodolfo Citro**[12], **Margot De Marco**[13], **Justin T. Tretter**[14], **Kim L. McBride**[15], **Dianna M. Milewicz**[1], **Simon C. Body**[16], **Siddharth K. Prakash**[1]*, **EBAV Investigators**[¶], **BAVCon Investigators**[¶]

1 University of Texas Health Science Center at Houston, Houston, Texas, United States of America, 2 University College Dublin School of Medicine, Dublin, Ireland, 3 Mayo Clinic, Rochester, Minnesota, United States of America, 4 Vall d'Hebron University Hospital, Barcelona, Spain, 5 Ghent University Hospital, Ghent, Belgium, 6 University of Nebraska Medical Center, Omaha, Nebraska, United States of America, 7 University of Leeds School of Medicine, Leeds, United Kingdom, 8 Consiglio Nazionale delle Richerche (CNR), Instituto di Fisiologia Clinica, Pisa, Italy, 9 University of Texas Health Science Center at San Antonio, San Antonio, Texas, United States of America, 10 Hoag Memorial Hospital Presbyterian, Newport Beach, California, United States of America, 11 Perelman School of Medicine at the University of Pennsylvania, Philadelphia, Pennsylvania, United States of America, 12 University Hospital "San Giovanni di Dio e Ruggi d'Aragona," Salerno, Italy, 13 Schola Medica Salernitana, University of Salerno, Baronissi, Italy, 14 Cleveland Clinic, Cleveland, Ohio, United States of America, 15 University of Calgary Cumming School of Medicine, Calgary, Alberta, Canada, 16 Boston University School of Medicine, Boston, Massachusetts, United States of America

¶ Membership of the EBAV and BAVCon Investigators are provided in the Acknowledgments.
* Siddharth.K.Prakash@uth.tmc.edu

**Data Availability Statement:** All relevant data are included in the manuscript or Supporting Information files. All genotype and CNV call data

## Abstract

Bicuspid aortic valve (BAV), the most common congenital heart defect, is a major cause of aortic valve disease requiring valve interventions and thoracic aortic aneurysms predisposing to acute aortic dissections. The spectrum of BAV ranges from early onset valve and aortic complications (EBAV) to sporadic late onset disease. Rare genomic copy number variants (CNVs) have previously been implicated in the development of BAV and thoracic aortic aneurysms. We determined the frequency and gene content of rare CNVs in EBAV probands (n = 272) using genome-wide SNP microarray analysis and three complementary CNV detection algorithms (cnvPartition, PennCNV, and QuantiSNP). Unselected control genotypes from the Database of Genotypes and Phenotypes were analyzed using identical methods. We filtered the data to select large genic CNVs that were detected by multiple algorithms. Findings were replicated in a BAV cohort with late onset sporadic disease (n = 5040). We identified 3 large and rare (< 1,1000 in controls) CNVs in EBAV probands. The burden of CNVs intersecting with genes known to cause BAV when mutated was increased in case-control analysis. CNVs intersecting with *GATA4* and *DSCAM* were enriched in cases, recurrent in other datasets, and segregated with disease in families. In total, we identified potentially pathogenic CNVs in 9% of EBAV cases, implicating alterations of candidate genes at these loci in the pathogenesis of BAV.

were uploaded to zenodo (10.5281/zenodo.12655912) and will be available from the Database of Genotypes and Phenotypes.

**Funding:** This study was supported in part by grants R01HL137028 (SP), R01HL114823 (SCB), and R21HL150373 (SCB) from the National Heart, Lung, and Blood Institute (NHLBI). The funder did not play any role in the study design, data collection, data analysis, decision to publish, or preparation of the manuscript.

**Competing interests:** The authors have declared that no competing interests exist.

## Introduction

Copy number variants (CNVs) have been implicated as causes or modifiers of many human diseases [1]. Specifically, large genomic CNVs are significantly enriched in cohorts with developmental delay or congenital abnormalities, and the severity of phenotypes has been correlated with the burden of rare CNVs [2]. These observations show that large, rare, *de novo* CNVs are likely to be pathogenic and can exert clinically relevant effects on disease pathogenesis [3, 4].

The worldwide prevalence of congenital heart disease (CHD) is 8.2 per 1000 live births [5]. CNVs have been implicated in both syndromic and non-syndromic forms of CHD [6–10]. The pathogenicity and penetrance of CNVs was initially established for clinical syndromes such as velocardiofacial syndrome, Turner syndrome, or Williams–Beuren syndrome, which involve chromosomal or megabase scale duplications or deletions, but has since been expanded to include additional CHD subtypes [10]. CNVs contribute to 10% of all CHD cases and up to 25% of cases with extracardiac anomalies or other syndromic features [11]. The role of pathogenic CNVs affecting genes that are known to cause CHD when mutated, such as *ELN* and *TBX1*, has been established [12]. Furthermore, population-level analysis has consistently demonstrated an increased burden of CHD in carriers of CNVs at specific genomic hotspots compared to controls, displaying the pathogenic potential of rare or *de novo* CNVs [12–14].

Bicuspid aortic valve (BAV) is the most common congenital heart malformation with a population prevalence of 0.5–2% [15]. BAV predisposes to aortic valve stenosis and thoracic aortic aneurysms and is associated with other left ventricular outflow tract lesions such as mitral valve disease and coarctation [16]. The high heritability of BAV was demonstrated in first- and second-degree relatives, who are more than ten times more likely to be diagnosed with BAV compared to matched controls [17]. BAV can occur as an isolated congenital lesion or as part of a clinical syndrome. For example, the prevalence of BAV is increased in Velocardiofacial, Loeys-Dietz, Kabuki, and Turner syndromes. Pathogenic variants of several genes are implicated in familial non-syndromic BAV, which is typically inherited as an autosomal dominant trait with reduced penetrance and variable expressivity. There is strong cumulative evidence that pathogenic variants in *GATA4*, *GATA6*, *MIB1*, *NOTCH1*, *ROBO4*, *SMAD4*, and *SMAD6* each contribute to a small percentage of non-syndromic BAV cases [18, 19]. Phenotypic expression of BAV disease ranges from incidental discovery in late adulthood to neonatal or childhood onset with severe manifestations requiring valve or aortic interventions. In comparison to patients with later disease onset, younger BAV cohorts tend to present with syndromic features or complex congenital malformations that are more likely to have a genetic cause, thereby increasing the power of association studies to discover clinically relevant CNVs [20]. Recently, we identified recurrent rare CNVs that were enriched for cardiac developmental genes in a young cohort with early-onset thoracic aortic aneurysms or acute aortic dissections [21].

We hypothesize that large rare genomic CNVs contribute to early onset complications of BAV. Consistent with previous observations, we predict that the burden and penetrance of rare CNVs will be increased in individuals with early onset disease when compared to elderly sporadic BAV cases and population controls. Identification of novel pathogenic CNVs can provide new insights into the genetic complexity of BAV and may be useful for personalized risk stratification or clinical guidance based on the specific recurrent CNV (Fig 1) [22]. Therefore, we set out to describe the burden and penetrance of rare CNVs in a young cohort with early onset complications of BAV disease (EBAV).

## Materials and methods

The study protocol was approved by the Committee for the Protection of Human Subjects at the University of Texas Health Science Center at Houston (HSC-MS-11-0185). Study

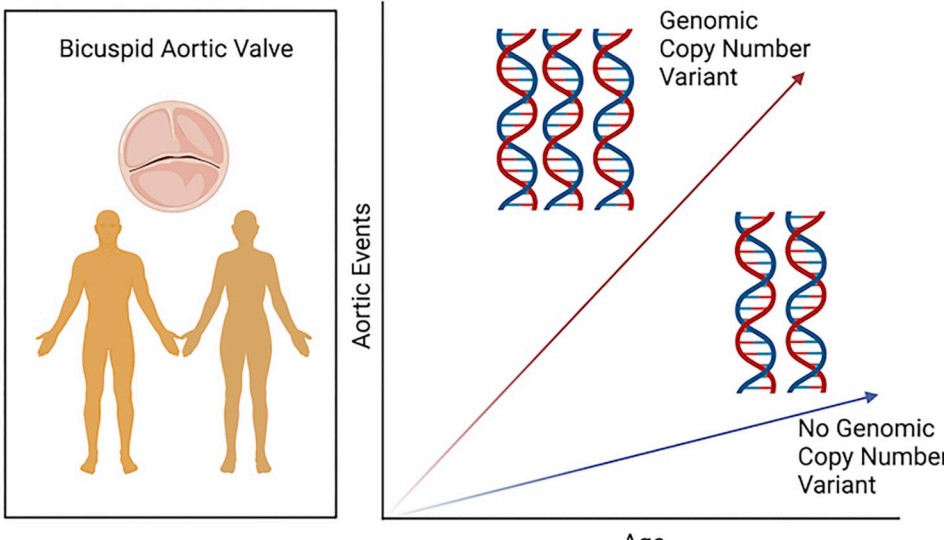

**Take-Home Message:** In adults with bicuspid aortic valve, specific rare genomic copy number variants may predict the onset and severity of valve or aortic complications.

**Fig 1. Specific rare genomic copy number variants may influence BAV disease severity.** Aortic events, thoracic aortic aneurysm, thoracic aortic dissection, or aortic valve stenosis or regurgitation requiring aortic valve repair or replacement.

recruitment began on July 1, 2017, and concluded on March 30, 2022. After written informed consent, we enrolled probands with early onset BAV disease (EBAV), which we defined as individuals with BAV who were under the age of 30 at the time of first clinical event. Clinical events were defined as aortic replacement, aortic valve surgery, aortic dissection, moderate or severe aortic stenosis or aortic regurgitation, large aneurysm (Z > 4.5), or intervention for BAV-related conditions. Those with hypoplastic left heart, known genetic mutations, genetic syndromes, or complex congenital heart disease were excluded. Samples were collected and genotyped as previously reported [23]. For comparison, we analyzed a cohort of older individuals of European ancestry with sporadic BAV disease selected from the International BAV Consortium (BAVGWAS) [24].

Phenotypes were derived from record review with confirmation of image data whenever possible [25, 26]. The computational pipeline for CNV analysis of Illumina single nucleotide polymorphism (SNP) array data included three independent CNV detection algorithms (Fig 2, S1 Appendix).

GenomeStudio was used to exclude samples with indeterminate sex or more than 5% missing genotypes, and single nucleotide polymorphisms (SNPs) with GenTrain = 0. Principal component analysis was used to remove outliers that did not cluster with European ancestry. Prior to CNV analysis, each dataset was trimmed by selecting a common set of 650,000 SNPs that were genotyped on each of the microarrays used in this study.

Three independent algorithms (PennCNV, cnvPartition, and QuantiSNP) were used to generate CNV calls and sample-level quality statistics from SNP intensity data. PennCNV and QuantiSNP were run on Unix clusters and cnvPartition data were exported from GenomeStudio. The analysis was run using default configurations.

PennCNV was used to generate QC data and remove CNV calls that intersect with polymorphic genomic regions. Samples that met any of the following criteria were excluded, standard deviation of the LogR ratio (obtained from PennCNV) > 0.35 or number of CNVs > 2

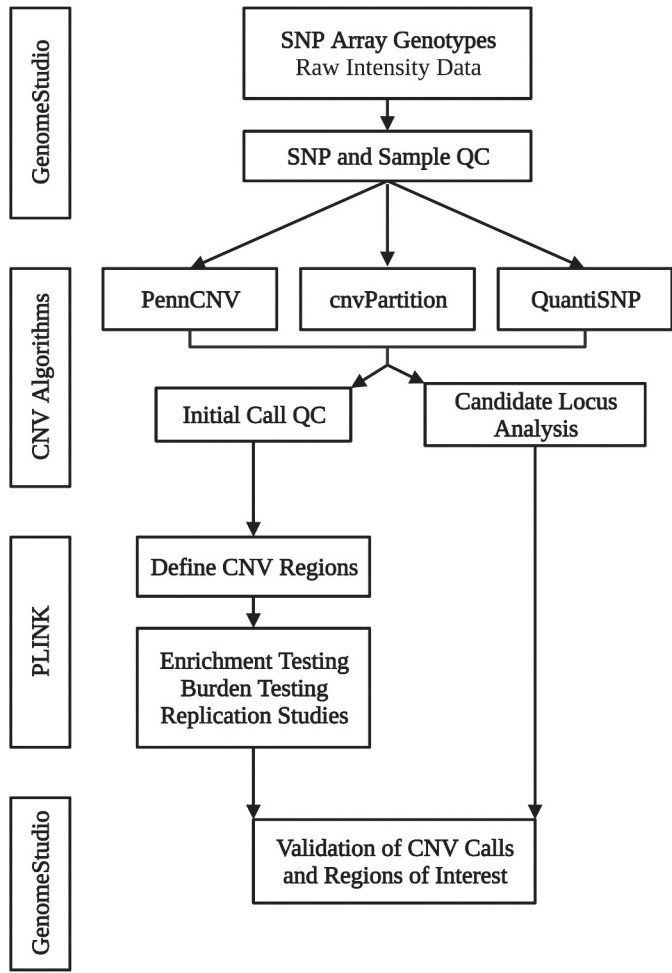

**Fig 2. Overview of pipeline for CNV identification and validation.** SNP, single nucleotide polymorphism; QC, Quality control; CNV, copy number variant. Illumina B-allele frequency and signal intensity data was trimmed and exported using GenomeStudio. Three different algorithms (PennCNV [27], cnvPartition, and QuantiSNP [28]) were used to generate initial CNV calls and sample-level statistics. Sample-level quality control analysis was performed using PennCNV. PLINK [29] was used to define CNV regions for subsequent burden, enrichment, and replication tests. Raw CNV calls were individually screened for CNVs intersecting with genes implicated in BAV and enriched in case-control tests. CNVs were validated by examining the raw data in GenomeStudio.

standard deviations above the mean for each data set. CNV calls less than 20 Kilobases in length and/or spanned by fewer than 6 probes were excluded. The overlap function for rare CNVs in PLINK was used to construct CNV regions (CNVRs) after adjacent regions were merged using PennCNV.

LogR ratio (LRR) and B allele frequency (BAF) data at CNVRs and calls of interest were visualized in GenomeStudio for validation. For segregation analysis, GenomeStudio was used to determine the presence of CNVs in relatives.

A total of 22,014 unselected control Illumina Genotypes obtained from the Database of Genotypes and Phenotypes were analyzed using identical methods (S1 Table). The Wisconsin Longitudinal Study (WLS) includes data on a cohort of 10,300 individuals who graduated from Wisconsin high schools in 1957. The Health and Retirement Study (HRS) includes data on 37,000 individuals aged 50 above from 23,000 households across the United States. Principal component analysis was used to select European ancestry genotypes from these datasets for

analysis. Datasets were paired for case-control analysis based on the concordance of log-transformed sample-level quality control statistics (number of CNV calls and standard deviation of logR ratios). Chi-squared or Fisher exact tests were used to compare CNV frequencies in cases and controls.

Rare CNV functions in PLINK (v1.7) were used to perform permutation-based burden tests or gene set-based enrichment tests as previously described [29, 30]. Case control burden tests were restricted to CNVs that were longer than 110 Kb and less than 0.1% in frequency. CNV overlap functions in PLINK were used to identify rare CNVs that intersect between datasets or involve specific BAV or CHD genes (S2 Table). The list of candidate genes included 190 CHD genes that have strong cumulative evidence to cause BAV or related congenital malformations from human or animal model data [31–33]. Genome Reference Consortium Human Build 37 was used for CNV annotation [34].

## Results

The EBAV cohort included a total of 544 samples, 272 EBAV probands, 21 relatives with BAV, and 251 apparently unaffected family members (26 trios and 15 multiplex families). The BAVGWAS sample contained 5,040 genotypes with associated demographic and clinical data. After exclusions due to data quality control or missing phenotypic data, 499 EBAV genotypes and 4,216 BAVGWAS were included in the final analysis. In phenotypic comparisons, EBAV probands were significantly younger at diagnosis, had more frequent co-existing congenital heart and vascular lesions, and underwent more frequent valve or aortic interventions (Table 1).

In comparisons between EBAV and WLS data, the rate of large CNVs was increased in EBAV cases compared to WLS controls, driven primarily by enrichment of large rare genomic deletions ($P<0.001$, Table 2, S3 Table).

In BAVGWAS cases, there were no significant differences in CNV rates or the proportions of individuals with large rare CNVs in comparison to HRS controls (Table 3).

We discovered 32 large (>250 kb) and rare (<1:1000) CNVs in EBAV probands, and 26 of these CNVs included protein-coding genes (S4 Table). The overall burden of large rare genic CNVs was not different between EBAV cases and WLS controls. However, the burden of large rare genic CNVs intersecting with genes known to cause BAV when mutated or implicated in syndromic BAV was significantly increased in EBAV cases (Table 4). EBAV probands with large rare CNVs were more likely to have concomitant congenital heart lesions and have other family members who were diagnosed with BAV (S5 Table).

**Table 1. Characteristics of EBAV and BAVGWAS probands.**

|  | EBAV (n = 272) | BAVGWAS (n = 3141) |
|---|---|---|
| **Female (%)** | 35 | 29 |
| **Age at diagnosis (years)** | 36 ± 21 | 52 ± 16 |
| **AA (%)** | 13 | 37 |
| **Predominant AR (%)** | 14 | 40 |
| **Predominant AS (%)** | 18 | 37 |
| **Other Lesions (%)** | 22 | 1 |
| **Aortic Valve Surgery (%)** | 33 | 16 |

n, number of cases; ±, standard deviation; AA, aortic aneurysm; AR, aortic regurgitation; AS, aortic stenosis; Other Lesions, other congenital heart malformations (primarily coarctation or ventricular septal defect).

**Table 2. Burden analysis of EBAV CNVs.**

|  | RATE | *P* | PROP | *P* | TOT | *P* | AVG | *P* |
|---|---|---|---|---|---|---|---|---|
| **Large** | 0.51 | <0.001 | 0.17 | 1 | 2648 | <0.001 | 690 | <0.001 |
| **Rare** | 0.36 | 0.79 | 0.21 | 1 | 426 | <0.001 | 288 | 0.04 |
| **Duplications** | 0.07 | 0.96 | 0.07 | 0.96 | 648 | 0.25 | 615 | 0.18 |
| **Deletions** | 0.11 | <0.001 | 0.05 | 0.02 | 1477 | 0.001 | 608 | 0.23 |

Large, CNV regions between 250 Kb and 5 Mb in length; Rare, occur in fewer than 1 in 1000 individuals; RATE, number of CNVs per individual; PROP, proportion of samples with one or more CNVs; TOT, total length of all CNVs in kilobases; AVG, mean CNV length; *P*, permuted *P*-value. Tests are 1-sided with 100,000 permutations.

In region-based association tests, 55 genes spanned by 26 CNVs were enriched in the EBAV cohort compared to WLS controls (S6 and S7 Tables). The largest CNVs involved *DSCAM* in 21q22 and GATA4 in 8p23. Large duplications involving the Velocardiofacial (VCFS) region in 22q11.2 and a recurrent CHD-associated CNV region in 1q21.1 were also enriched in EBAV probands. In BAVGWAS cases, common CNVs involving *NANOG* in 12p13.31 and rare CNVs involving *NIBPL* in 5p13.2, which are essential for early heart development, are enriched in comparison to HRS controls (S8 and S9 Tables). *NIBPL* mutations cause Cornelia-de Lange syndrome with a spectrum of congenital heart malformations including BAV.

We also scrutinized candidate genomic regions that are implicated in CHD by analyzing data from individual CNV algorithms to detect copy number alterations that may have been filtered out due to strict quality control criteria prior to enrichment studies. We identified additional rare EBAV CNVs that intersect with CHD candidate genes *CELSR1*, *GJA5*, *RAF1*, *LTBP1*, *KIF1A*, *MYH11*, *TTN*, and the VCFS region in 22q11.2. We detected additional *GATA4* and *DSCAM* CNVs in multiplex families. These CNVs were enriched in EBAV cases compared to WLS controls (Table 5, S10 Table). In total, 9% of EBAV probands had CNVs that are likely to contribute to the development of BAV (S11 Table).

We similarly scrutinized BAVGWAS data for additional CNVs that intersect with CHD genes. We found that large duplications involving *SOX7* and *GATA4* in 8p23 and the VCFS region in 22q11.2 were also significantly enriched in BAVGWAS cases compared to HRS controls (Table 6, S12 Table).

Next, we attempted to replicate our observations by identifying CNVs in the BAVGWAS dataset that overlapped with EBAV CNVs. Large rare CNVs intersecting with *GATA4*, *DSCAM*, *CELSR1*, *GJA5*, *MYH11*, *KIF1A*, and *TBX1* overlapped between the EBAV and BAVGWAS datasets (S13 Table). However, only CNVs intersecting with *GATA4* and *DSCAM* were significantly enriched in both datasets (Fig 3).

**Table 3. Burden analysis of BAVGWAS CNVs.**

|  | RATE | *P* | PROP | *P* | TOT | *P* | AVG | *P* |
|---|---|---|---|---|---|---|---|---|
| **Large** | 0.23 | 1 | 0.20 | 1 | 688 | 0.02 | 581 | <0.001 |
| **Rare** | 0.28 | 1 | 0.23 | 1 | 306 | 0.87 | 253 | 0.6 |
| **Duplications** | 0.18 | 1 | 0.15 | 1 | 309 | 0.98 | 266 | 0.98 |
| **Deletions** | 0.11 | 1 | 0.10 | 1 | 244 | 0.04 | 226 | 0.02 |

Large, CNV regions between 250 Kb and 5 Mb in length; Rare, occur in fewer than 1 in 1000 individuals; RATE, number of CNVs per individual; PROP, proportion of samples with one or more CNVs; TOT, total length of all CNVs in kilobases; AVG, mean CNV length; *P*, permuted *P* value. Tests are 1-sided with 100,000 permutations.

**Table 4. Burden of rare EBAV CNVs.**

| | EBAV | | WLS | | | |
|---|---|---|---|---|---|---|
| | **Calls** | **Rate** | **Calls** | **Rate** | **RR** | ***P*** |
| **Genic** | 26 | 0.84 | 1151 | 0.65 | 1.2 | 0.18 |
| **Deletions** | 10 | 0.04 | 439 | 0.05 | 0.75 | 0.85 |
| **BAV** | 4 | 0.01 | 0 | 0.00 | 296 | $1 \times 10^{-5}$ |

Calls, total number of CNVs that met the specified criteria; Rate, number of CNVs per individual; RR, relative risk; *P*, *P* value; Genic, CNVs that intersect with genes; BAV, CNVs that intersect with genes that are known to cause bicuspid aortic valve (BAV) when mutated or implicated in syndromic BAV; Total, total number of large, rare CNVs. *P* values were calculated using 100,000 permutations.

We also identified 21 very large genomic CNVs > 5 Mb in length in the BAVGWAS dataset. Analysis of GenomeStudio data showed that most of these were mosaic loss of heterozygosity regions or duplications. Nine were large germline chromosome-scale aberrations, including two cases of trisomy 21 (S14 Table). We did not identify any large X chromosome copy variants that may be consistent with Turner syndrome. There were no megabase-scale copy number variants in the EBAV dataset.

Pedigree analysis showed that CNVs involving *CELSR1*, *LTBP1*, *KIF1A*, *GATA4*, and *DSCAM* segregate with BAV in EBAV families (S1 Fig). Most of these CNVs occurred *de novo* in probands and were not found in unaffected family members. CNV carriers tended to present due to moderate or severe aortic regurgitation requiring valvular surgery. One proband had aortic coarctation. The age at presentation or sex of individuals with rare CNVs was not significantly different from the rest of the EBAV cohort.

## Discussion

We identified large, rare, and likely pathogenic CNVs in almost 10% of EBAV probands that are enriched in genes that cause BAV when mutated. The percentage of EBAV cases with likely pathogenic CNVs is similar to our previous observations in a cohort with early onset TAD [36]. Enrichment of CNVs involving *GATA4* and *DSCAM* in EBAV cases replicated in two additional BAV datasets and thousands of unselected control genotypes. This analysis provides compelling evidence that rare CNVs collectively contribute to more BAV cases than any single mutated gene.

**Table 5. CNVs affecting congenital heart disease genes in EBAV.**

| Region | Genes | Case | Control | OR | P | 95% CI |
|---|---|---|---|---|---|---|
| 22:46261909–51187440 | *CELSR1* | 1 | 1 | 33 | 0.07 | 2.1 to 530 |
| 1:146326373–147340734 | *GJA5* | 1 | 2 | 17 | 0.17 | 1.5 to 183 |
| 3:12599717–12803792 | *RAF1* | 1 | 2 | 17 | 0.16 | 1.5 to 183 |
| 22:41278694–41813285 | *DSCAM* | 4 | 2 | 67 | <0.001 | 12 to 367 |
| 8:11495032–11856903 | *GATA4* | 4 | 0 | 301 | <0.001 | 16 to 5599 |
| 22:19000000–22000000 | *TBX1, CRKL* | 4 | 10 | 13 | <0.001 | 4.2 to 43 |
| 16:15484868–16295863 | *MYH11* | 2 | 22 | 3.0 | 0.34 | 0.70 to 13 |
| 2:241652252–241678528 | *KIF1A* | 3 | 22 | 4.5 | 0.04 | 1.3 to 15 |
| 2:32775984–33331219 | *LTBP1* | 2 | 26 | 2.5 | 0.45 | 0.60 to 11 |

Region, hg38 coordinates corresponding to the minimum overlap region of CNVs; Genes, candidate genes in region; Case, number of rare CNVs in EBAV cases that intersect with region; Control, number of CNVs in WLS controls that intersect with region; OR, odds ratio; *P*, chi-squared *P*-value; 95% CI, 95% confidence interval. Inherited CNVs were only counted once.

**Table 6. CNVs affecting congenital heart disease genes in BAVGWAS.**

| Region | Genes | Case | Control | OR | P | 95% CI |
|---|---|---|---|---|---|---|
| 3:29993977–31273870 | *TGFBR2* | 1 | 0 | 5.6 | 0.75 | 0.23 to 138 |
| 9:101861767–102092282 | *TGFBR1* | 1 | 0 | 5.6 | 0.75 | 0.23 to 138 |
| 21:41577819–41842252 | *DSCAM* | 2 | 1 | 3.7 | 0.58 | 0.34 to 41 |
| 22:46924254–46931077 | *CELSR1* | 3 | 1 | 5.6 | 0.25 | 0.58 to 54 |
| 2:111404636–11310378 | *TMEM87B, FBLN7* | 3 | 2 | 2.8 | 0.48 | 0.47 to 17 |
| 8:11385469–11821835 | *GATA4* | 8 | 1 | 15 | 0.002 | 1.9 to 120 |
| 2:147166377–147308112 | *GJA5* | 4 | 10 | 0.75 | 0.83 | 0.23 to 2.4 |
| 16:29664753–30199713 | *MAPK3* | 3 | 15 | 0.37 | 0.17 | 0.11 to 1.3 |
| 22:19000000–22000000 | *TBX1, CRKL* | 18 | 11 | 3.1 | 0.004 | 1.4 to 6.5 |
| 2:32689829–33299434 | *LTBP1* | 9 | 22 | 0.76 | 0.62 | 0.35 to 1.7 |
| 16:15240816–16281154 | *MYH11* | 13 | 27 | 0.90 | 0.89 | 0.46 to 1.7 |
| 2:241640262–241689833 | *KIF1A* | 13 | 30 | 0.81 | 0.64 | 0.42 to 1.6 |

Region, hg38 coordinates corresponding to the minimum overlap region of CNVs; Genes, candidate genes in region; Case, number of rare CNVs in EBAV cases that intersect with region; Control, number of CNVs in WLS controls that intersect with region; OR, odds ratio; *P*, chi-squared *P*-value; 95% CI, 95% confidence interval. Inherited CNVs were only counted once.

GATA-binding protein 4 (*GATA4*) is a transcription factor that is required for cardiac and neuronal differentiation during embryogenesis [37]. Mutations of *GATA4* and its homologs *GATA5* and *GATA6* cause congenital heart lesions [38]. Mutations in the *GATA4* gene have been linked to a range of congenital heart diseases in humans, such as cardiac septal defects, tetralogy of Fallot, and patent ductus arteriosus [39]. Patients with BAV who have rare functional variants in the *GATA* family exhibit varying degrees of aortopathy expression, including aortic aneurysm, dissection, and/or aortic stenosis. Alonso-Montes et al. described 4 predicted deleterious *GATA4* mutations in 122 non-syndromic BAV probands who did not have affected relatives [40]. Rare *GATA4* deletions and putative loss of function mutations are also implicated in CHD with distinctive features, underlining the importance of *GATA4* dosage to cardiac development [41, 42]. Glessner et al. discovered large *de novo* duplications involving *GATA4* in CHD trios with conotruncal defects or left ventricular outflow tract obstructive lesions [43]. Some duplications were inherited from apparently unaffected parents. Zogopoulos and Yu described similar genomic duplications in unaffected individuals and in unselected control genotypes [44, 45].

These observations are consistent with low-penetrance CHD in *GATA4* duplication carriers. Similar to other complex and multifactorial disorders, CHD pathogenesis is likely caused by the cumulative impact of multiple CNVs or mutations, each exerting small to moderate effects to collectively disrupt cardiac development. For example, the frequency of congenital heart lesions is increased in individuals who have both 22q11.2 deletions and a common 12p13.31 duplication involving the *SLC2A3* gene. The *SLC2A3* CNV likely functions as a modifier of the cardiac phenotype associated with 22q11 deletion syndrome, exemplifying a "two-hit" model [46].

More than half of patients with Down syndrome have congenital heart malformations due to the interaction of multiple dosage-sensitive CHD genes on chromosome 21 [47–49]. Down syndrome cell adhesion molecule (*DSCAM*), previously shown to play a critical role in neurogenesis, has also been implicated in the pathophysiology of CHD [50]. Analysis of rare segmental trisomies of chromosome 21 suggested that duplication of *DSCAM* and the contiguous *COL6A1* and *COL6A2* genes may cause septal abnormalities and other Down Syndrome-

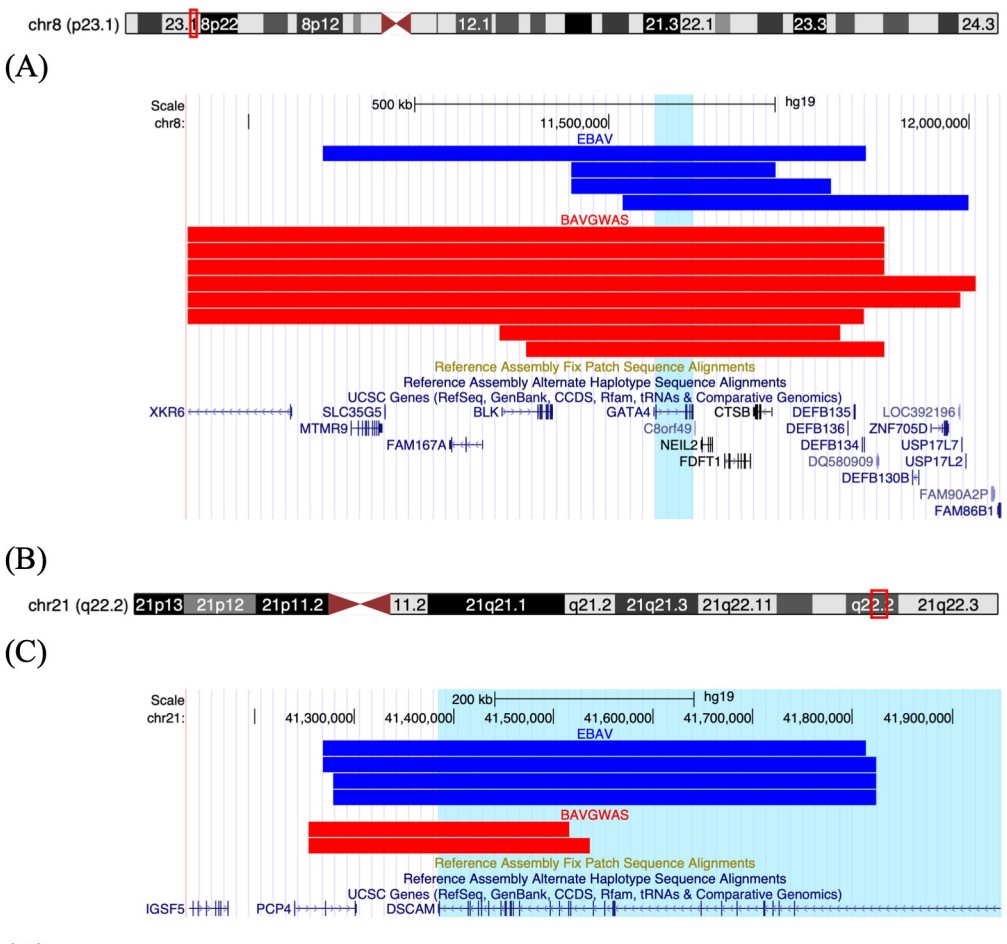

**Fig 3. UCSC genome browser plots of *GATA4* and *DSCAM* variants.** Each bar represents a copy number variant (CNV). Blue, EBAV CNVs; Red: CNVs BAVGWAS CNVs. (a) Ideogram of Chromosome 8 with CNV region highlighted red; (b) Plot of *GATA4* CNVs; (c) Ideogram of Chromosome 21 with CNV region highlighted red; (d) Plot of *DSCAM* CNVs. Figures were constructed using the UCSC Genome Browser (http://genome.ucsc.edu) [35].

related CHD lesions, including BAV. Overexpression of *DSCAM* and *COL6A2* causes cardiac malformations in mice [51]. Our findings suggest that rare CNVs involving *DSCAM* may contribute to some non-syndromic BAV cases.

*GATA4* and *DSCAM* CNVs segregated with disease in multiple families but are not fully penetrant and were detected in some unaffected relatives. Intriguingly, large 22q11.2, *GATA4* and *DSCAM* CNVs were more highly enriched in EBAV than in BAVGWAS cases, suggesting that these CNVs may drive early onset BAV disease. These results are consistent with our observation that pathogenic CNVs involving candidate BAV genes are also enriched in EBAV compared to BAVGWAS cases. Our data suggests that pathogenic CNVs at these loci may predict accelerated disease onset or more severe complications.

We also identified recurrent rare CNVs involving specific dosage-sensitive cardiac developmental genes that are implicated in non-syndromic CHD. Recurrent 1q21.1 distal deletions encompassing *GJA5*, the gene encoding Connexin-40, are associated with CHD lesions including BAV. Enrichment of small genomic duplications spanning the *GJA5* gene in cohorts with tetralogy of Fallot and cardiac abnormalities in mice with a targeted *GJA5* deletion imply that

dosage variations of *GJA5* contribute to CHD [52]. *CELSR1*, a cadherin superfamily member, is mutated in families with BAV and hypoplastic left heart syndrome [53]. *LTBP1* encodes an extracellular matrix protein that regulates TGF-β and fibrillin and has been implicated in congenital heart lesions [54]. *KIF1A*, encoding a kinesin microtubule transporter, was implicated in a dominant multisystem syndromic disorder with valvular and cardiac defects [55]. Mutation of *MYH11* causes familial thoracic aortic aneurysms and dissections with an increased prevalence of BAV [56]. *TTN* mutations cause dilated cardiomyopathy and are associated with other left-sided congenital lesions [57]. Mutations or copy number changes involving these genes all cause a wide spectrum of penetrance and phenotypic severity, consistent with sensitivity to genetic or clinical modifiers.

Our combinatorial analysis method eliminated many CNVs that were detected by single algorithms or did not meet quality control benchmarks. Therefore, our analysis likely underestimated the contribution of rare pathogenic CNVs to BAV. We also recognize that cardiac development involves the complex interaction of many genes. We selectively validated individual CNVs at loci of interest but may have underrepresented CNVs that had no *a priori* relationship with CHD. The apparent penetrance of some CNVs may be less than expected due to missing phenotypic information. The available clinical data was not sufficiently detailed to permit genotype-phenotype correlations with specific CHD clinical features.

In conclusion, we identified large rare CNVs in a significant proportion of BAV cases, including a subset of CNVs that may predict early onset complications of BAV disease. These observations add to the evidence that rare CNVs may eventually have clinical utility for risk stratification and personalized disease management (Fig 1).

## Supporting information

**S1 Appendix. Computational pipeline for CNV analysis.**
(DOCX)

**S1 Table. Summary of control cohorts.** WLS, Wisconsin Longitudinal Study on Aging; HRS, Health and Retirement Study; Accession, accession number in the Database of Genotypes Phenotypes. WLS includes data on a cohort of 10,300 individuals who graduated from Wisconsin high schools in 1957. HRS includes data on 37,000 individuals aged 50 above from 23,000 households across the United States.
(DOCX)

**S2 Table. List of genes implicated in BAV or CHD.** Used for intersection studies.
(DOCX)

**S3 Table. Summary of CNV data.** EBAV, early onset bicuspid aortic valve cohort; BAVG-WAS, Genome-wide Association Study from the International BAV Consortium; WLS, Wisconsin Longitudinal Study on Aging; HRS, Health Retirement Study; PennCNV, number of CNV calls detected by PennCNV algorithm after quality control; cnvPartition, number of CNV calls detected by cnvPartition algorithm after quality control; QuantiSNP, number of CNVs detected by QuantiSNP algorithm after quality control; Merged, number of CNV regions after merging adjacent calls; >5 MB, number of CNV regions that are larger than 5 megabases; Rare, number of CNVs that occur in less than 1 in 1000 samples of the combined datasets; Rare Deletions, number of large (> 250 Kb) rare deletions.
(DOCX)

**S4 Table. Large rare copy number variants in the EBAV probands.** Gene(s), genes intersected by CNV; Chr, chromosome; Start BP, start base pair of CNV; Stop BP, stop base pair of

CNV; DUP, duplication; DEL, deletion. All CNVs were validated by direct inspection in GenomeStudio.
(DOCX)

**S5 Table. Phenotype comparison of EBAV samples with without large, rare CNVs.** CNV, samples with large rare CNVs; n, number of samples; TAA, thoracic aortic aneurysm; AR, aortic regurgitation; AS, aortic stenosis; Other lesions, other congenital heart malformations; >1 Affected, number of families with more than one affected individual. Percentages are in parentheses. *Significantly increased.
(DOCX)

**S6 Table. Top hits in region-based association tests of EBAV CNVs compared to WLS controls.** Chr, Chromosome; EMP1, permutation-based empiric *P*-value; EMP2, after genome-wide correction. *Top candidate genes.
(DOCX)

**S7 Table. Rare CNVs enriched in EBAV cohort.** Gene(s), genes intersected by CNV; Chr, chromosome; Start, start base pair of CNV; Stop, stop base pair of CNV; DUP, duplication; DEL, deletion. *Call in apparently unaffected family member. **Call in affected family member belonging to multiplex family.
(DOCX)

**S8 Table. Top hits in region-based association tests of BAVGWAS CNVs compared to HRS controls.** Chr, Chromosome; EMP1, permutation-based empiric *P*-value; EMP2, after genome-wide correction. *Top candidate genes.
(DOCX)

**S9 Table. Rare CNVs enriched in BAVGWAS cohort.** Gene(s), genes intersected by CNV; Chr, chromosome; Start, start base pair of CNV; Stop, stop base pair of CNV; DUP, duplication; DEL, deletion.
(DOCX)

**S10 Table. EBAV CNVs intersecting with congenital heart disease genes.** Chr, chromosome; Start, start base pair of CNV; Stop, stop base pair of CNV; DUP, duplication; DEL, deletion. *Call in unaffected family member, **Call in affected family member from a multiplex family.
(DOCX)

**S11 Table. Phenotype information of EBAV probands with candidate rare CNVs.** Gene, principal gene/region intersected by CNV; TAA, thoracic aortic aneurysm, TAD, thoracic aortic dissection; AS, aortic stenosis; AR, aortic regurgitation; AVR, aortic valve replacement; Aortic Repair, open or endovascular aortic procedure.
(DOCX)

**S12 Table. BAVGWAS CNVs intersecting with congenital heart disease genes.** Chr, chromosome; Start, start base pair of CNV; Stop, stop base pair of CNV; DUP, duplication; DEL, deletion.
(DOCX)

**S13 Table. CNV overlaps between EBAV and BAVGWAS cohorts.** Chr, chromosome; Start, start base pair of CNV; Stop, stop base pair of CNV; DUP, duplication; DEL, deletion. Some *GATA4* CNVs were not identified in the overlap analysis because they exceeded the size

threshold (5 Mb).
(DOCX)

**S14 Table. Large genomic events in the BAVGWAS dataset.** Chr, chromosome; Start, start base pair of CNV; Stop, stop base pair of CNV; DUP, duplication; DEL, deletion; LOH, loss of heterozygosity.
(DOCX)

**S1 Fig. Segregation of candidate BAV CNVs.** (A) *GATA4* CNV; (B) *DSCAM* CNV; (C) *CELSR1* CNV; (D, E) *KIF1A* CNVs; (F) *LTBP1* CNVs. Dot, apparently unaffected CNV carrier; Shaded, affected CNV carrier; U, no genotype was available.
(TIFF)

**S1 File.**
(PNG)

## Acknowledgments

We thank Joana Castillo and Jacqueline Jennings for sample preparation, William J. Allen for computational support, and Gladys Zapata, Nitesh Mehta, and the Laboratory for Translational Genomics at Baylor College of Medicine for microarray genotyping. Figs 1 and 2 were created using BioRender.com. S1 Fig was created using CeGaT Pedigree Chart Designer. The Texas Advanced Computing Center (TACC) at The University of Texas at Austin (http://www.tacc.utexas.edu) provided high-performance computing resources for data analysis.

The EBAV Investigators are: Siddharth K. Prakash, Dianna M. Milewicz, Shaine A. Morris, Rita Milewski, Giuseppe Limongelli, Allesandro Della Corte, Laura Perrone, Yuli Y. Kim, Hector I. Michelena, Maria G. Andreassi, Arturo Evangelista, Denver Sallee, Angela Yetman, Kim McBride, Eduardo Bossone, Rodolfo Citro, Dawn S. Hui, Malenka M. Bissell, Andrea Ballotti, Ilenia Foffa, Margot De Marco, Anthony Caffarelli, Rita Weise, Julie DeBacker, Laura Muino Mosquera, Robbin Cohen, Laura Dos Subira, Justin T. Tretter, Anna Sabe Rotes, Martina Caiazza, Lamia Ait Ali, Francesca Pluchinotta, Simon C. Body.

Lead author: Siddharth K. Prakash
Siddharth.K.Prakash@uth.tmc.edu

The BAVCon Investigators are: Simon C. Body, Alessandro Della Corte, Rodolfo Citro, Yohan Bossé, Alessandro Frigiola, Andrea Ballotta, Arturo Evangelista, Evaldas Girdsaukas, Betti Giusti, Bo Yang, Carlo de Vincentiis, Dan Gilon, Thoralf M. Sundt, David Messika Zeitoun, Dianna M. Milewicz, Siddharth K. Prakash, Eduardo Bossone, Eric Eisselbacher, Vicenza Stefano Nistri, Francesca R. Pluchinotta, Giuseppe Limongelli, Gordon S. Huggins, Joshua C. Denny, Patrick M. McCarthy, S. Chris Malaisrie, Aakash Bavishi, Hector I. Michelena, J. Daniel Muehlschelgel, Kim Eagle, Lasse Folkersen, Malenka M. Bissell, Patrick Mathieu, Per Eriksson, Peter Lichtner, Ronen Durst, Sébastien Thériault.

Lead author: Simon C. Body
scbody@bu.edu

## Author Contributions

**Conceptualization:** Siddharth K. Prakash.

**Data curation:** Hector I. Michelena, Anna Sabate-Rotes, Lisa Bianco, Julie De Backer, Laura Muiño Mosquera, Anji T. Yetman, Malenka M. Bissell, Maria Grazia Andreassi, Ilenia Foffa, Dawn S. Hui, Anthony Caffarelli, Yuli Y. Kim, Dongchuan Guo, Rodolfo Citro,

Margot De Marco, Justin T. Tretter, Kim L. McBride, Dianna M. Milewicz, Simon C. Body, Siddharth K. Prakash.

**Formal analysis:** Steven G. Carlisle, Hasan Albasha, Siddharth K. Prakash.

**Funding acquisition:** Siddharth K. Prakash.

**Methodology:** Steven G. Carlisle, Hasan Albasha, Simon C. Body, Siddharth K. Prakash.

**Project administration:** Dianna M. Milewicz, Simon C. Body, Siddharth K. Prakash.

**Resources:** Dianna M. Milewicz.

**Supervision:** Siddharth K. Prakash.

**Writing – original draft:** Steven G. Carlisle, Hasan Albasha, Siddharth K. Prakash.

**Writing – review & editing:** Steven G. Carlisle, Hasan Albasha, Julie De Backer, Laura Muiño Mosquera, Anji T. Yetman, Dongchuan Guo, Dianna M. Milewicz, Simon C. Body, Siddharth K. Prakash.

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
