## [Decision Letter · Decision Letter 0]

25 Jan 2024

PONE-D-23-37920Rare Genomic Copy Number Variants Implicate New Candidate Genes for Bicuspid Aortic ValvePLOS ONE

Dear Dr. Prakash, Thank you for submitting your manuscript to PLOS ONE. After careful consideration, we feel that it has merit but does not fully meet PLOS ONE’s publication criteria as it currently stands. Therefore, we invite you to submit a revised version of the manuscript that addresses the points raised during the review process.

We look forward to receiving your revised manuscript.

Kind regards,

Klaus Brusgaard

Academic Editor

PLOS ONE

Journal Requirements:

4. Please expand the acronym “NHLBI” (as indicated in your financial disclosure) so that it states the name of your funders in full.

"We thank Joana Castillo and Jacqueline Jennings for sample preparation, and Gladys Zapata, 

Nitesh Mehta, and the Laboratory for Translational Genomics at Baylor College of Medicine for 

microarray genotyping. This study was supported in part by R01HL137028 (SP). Fig 1 created 

with BioRender.com."

"SP R01HL135728 NHLBI. The funders did not play any role in the study design, data collection, data analysis, decision to publish, or preparation of the manuscript"

7. Two of the noted authors are group or consortium [EBAV Investigators and BAVCon Investigators]. In addition to naming the author group, please list the individual authors and affiliations within this group in the acknowledgments section of your manuscript. Please also indicate clearly a lead author for this group along with a contact email address.

Additional Editor Comments:

The study is relevant and of interest. The reviewers are in agreement that the mauscript at a number of points needs clarification and generally needs some word processing. Plieas pay attention to reviewer comments and adhere to these.

Reviewers' comments:

Reviewer's Responses to Questions

Comments to the Author

1. Is the manuscript technically sound, and do the data support the conclusions?

Reviewer #1: Yes

Reviewer #2: Partly

Reviewer #3: Yes

2. Has the statistical analysis been performed appropriately and rigorously?

Reviewer #1: No

Reviewer #2: I Don't Know

Reviewer #3: Yes

3. Have the authors made all data underlying the findings in their manuscript fully available?

Reviewer #1: Yes

Reviewer #2: Yes

Reviewer #3: Yes

4. Is the manuscript presented in an intelligible fashion and written in standard English?

Reviewer #1: Yes

Reviewer #2: Yes

Reviewer #3: Yes

5. Review Comments to the Author

Reviewer #1: The paper by Carlisle et al describes rare copy number variants in two sets of patients with bicuspid aortic valves. While this type of investigation has been described before, the paper is of interest as it has two different cohorts, one early onset and one late onset, and describes the differences and overlap between these findings. However, because of the two patient cohorts, multiple control cohorts and the differences between these cohorts, the paper could do with some textual cleaning up, as at times it is confusing.

The use of control cohorts to compare against needs more detail. WLS and HRS and Illumina Genotypes are used to compare CNVs against. What are these cohorts? Basic information on this is needed. Why are certain control cohorts used to compare against certain patient cohorts? Why not use one cohort, etc etc. How sure are the authors that these cohorts do not also contain patients with BAV (as BAV is a rather frequent phenotype), and would this be a problem?

The authors do three comparisons, EBAV alone, BAVGWAS alone and overlap between the two. However this is not quite clear from the results and the way it is written. I would like a clear separation between these comparisons in both text and tables, as that makes it much easier to follow. Is there a table in which the overlap between the two sets is described?

Table S9 is unclear. How can there be segregation when none of the family have the CNV? Eg BAV787. I would suggest including pedigrees, as that makes this type of information much easier to parse.

On line 191/193 prevalence is discussed, but I don’t see any statistics here to compare between the two groups.

On line 98 references are missing to the various papers that investigated the genes mentioned.

In the discussion, before line 335 there is a paragraph on GJA5 deletions , after this there is just a summary genes, each on their own sentence. This reads rather jarring in comparison to the previous part. I would put a connecting or explanatory sentence here, to prevent it reading like a summary.

Reviewer #2: The manuscript reports the association of copy number variants with bicuspid aortic valve disease. The authors report that CNVs could explain a notable number of cases. This is a very interesting study and advances genetic discovery efforts for BAV. I did however find some areas of the manuscript difficult to interpret. I believe the manuscript could be improved with a clearer description of findings. I list several points below.

Cohort numbers are confusingly presented. For EBAV 272 probands are mentioned in the abstract, 293 families on Page 6, a cohort size of 544 (comprising both affected and unaffected samples) presented in Table 1 and 279 probands in Table 2. For BAVGWAS the text refers to 5040, but Table 2 summarises just 3141. It is therefore unclear how many samples have contributed to the analyses. Can the authors please explain and address this so that numbers analysed are clearly presented.

I don’t see the need for Table 1 with the information included.

Page 9, paragraph 2. Presentation of Tables 3 and S2 in relation to the text could be improved. Is the “prevalence of large and rare CNVs”, referring to the PE for number of CNVs per individual?

Table S3 could include a column listing affected genes.

Where is the data supporting the page 5 statement “Seven of these genic CNVs were

205 enriched in EBAV cases compared to WLS controls with a genome-wide adjusted empiric P < 0.05.”

Which BAV genes are in the CNVs assessed in Table 4?

Page 10. “Large duplications involving the 208  Velocardiofacial (VCFS) region in 22q11.2 and 1q21.1 microduplications were also enriched in 209  EBAV cases (Table S4)”.

The table does not present evidence for enrichment.

Page 10. Which CHD genes were scrutinized? A supplementary table listing genes assessed would be helpful. I think important to show CHD associated genes where no CNV was identified.

Page 10, Table 5 and table S6. I believe “subtle” should be properly defined to clarify criteria of the CNVs analysed in this section. Relationship of the two tables is unclear – for some genes (e.g., MYH11) the CNVs described in Table S6 appear to correspond to the numbers presented in Table 5, however, this is not true for others (e.g., GATA4). Please explain.

Burden testing P-values should be presented to the data in Table 5 and Table 6.

Burden testing method is not properly described.

Abstract and Page 13, opening paragraph of discussion. The authors report potentially pathogenic variants for 8% of BAV cases in their EBAV cohort. It is not immediately clear to me which CNVs were considered as likely pathogenic. A table summarising the cases where CNVs have been interpreted in this way, with a clinical description of proband/family could be included.

The methods should include a section clearly describing the statistical tests performed.

Reviewer #3: The authors analyzed the frequency and genetic relevance of large and rare CNVs in a cohort of early onset bicuspid aortic valve disease. BAV is the most common congenital heart defect and a major cause of severe cardiac complications (early) or later in life. As the genetic causes of the disease are still incompletely understood and especially risk stratification is unclear the study is of great clinical interest. The genetic methods are appropriate, and the manuscript is well written.

However, I have some comments:

1. Did the authors check whether large and rare CNVs tend to occur more often in patients with BAV developing an aortic aneurysm or is the distribution equal?

2. Where the CNVs more common in families with more than 1 affected individual?

3. Interestingly, CNVs only affected the TGFB family in late onset BAV. How do the authors explain this finding?

3, The average clinical reader may not be used to CNVs. I therefore recommend including a simple graphical abstract.

6. PLOS authors have the option to publish the peer review history of their article (what does this mean?). If published, this will include your full peer review and any attached files.

Do you want your identity to be public for this peer review? For information about this choice, including consent withdrawal, please see our Privacy Policy.

Reviewer #1: No

Reviewer #2: No

Reviewer #3: No

---

## [Author Response · Author response to Decision Letter 0]

1 Mar 2024

Reviewer 1

1. The use of control cohorts to compare against needs more detail. WLS and HRS and Illumina Genotypes are used to compare CNVs against. What are these cohorts? Basic information on this is needed. Why are certain control cohorts used to compare against certain patient cohorts? Why not use one cohort, etc etc. How sure are the authors that these cohorts do not also contain patients with BAV (as BAV is a rather frequent phenotype), and would this be a problem?

We added a description of the control cohorts to S1 Table and lines 150-155: “The Wisconsin Longitudinal Study (WLS) includes data on a cohort of 10,300 individuals who graduated from Wisconsin high schools in 1957. The Health and Retirement Study (HRS) includes data on 37,000 individuals aged 50 above from 23,000 households across the United States. Principal component analysis was used to select European ancestry genotypes from these datasets for analysis.” We also added the rationale for selection of comparison cohorts to lines 155-157: “Datasets were paired for case-control analysis based on the concordance of log-transformed sample-level quality control statistics (number of CNV calls and standard deviation of the LogR Ratio).” While we acknowledge that unselected population cohorts may contain rare individuals with BAV, the effect, if present, would decrease the strength of reported associations (type II error).

2. The authors do three comparisons, EBAV alone, BAVGWAS alone and overlap between the two. However, this is not quite clear from the results and the way it is written. I would like a clear separation between these comparisons in both text and tables, as that makes it much easier to follow. Is there a table in which the overlap between the two sets is described?

We revised the Results section to separate EBAV and BAVGWAS results as requested. Burden analysis of the entire EBAV cohort is described in lines 181-184, followed by Table 2. Burden analysis of the BAVGWAS cohort is described in lines 192-193, followed by Table 3. Identification and enrichment analysis of large rare CNVs in the EBAV dataset is described in lines 201-218. Enrichment analysis of the BAVGWAS dataset is described in lines 218-222. We describe the overlap of large rare EBAV and BAVGWAS CNVs in lines 247-252. All overlapping rare CNVs are listed in Table S12.

3. Table S9 is unclear. How can there be segregation when none of the other family have the CNV? e.g. BAV787. I would suggest including pedigrees, as that makes this type of information much easier to parse.

We added a sentence in lines 266-267 to explain that these CNVs occurred as de novo in EBAV families. We replaced the supplemental table by S1 Figure that illustrates the pedigrees.

4. On line 191/193 prevalence is discussed, but I don’t see any statistics here to compare between the two groups.

We changed lines 181-183 as follows: “In comparisons between EBAV and WLS data, the rate of large CNVs was increased in EBAV cases compared to WLS controls, driven primarily by enrichment of large rare genomic deletions (P<0.001, Table 2).”

5. On line 98 references are missing to the various papers that investigated the genes mentioned.

We added two citations to line 81 that provide essential summary data about the genes in question.

6. In the discussion, before line 335 there is a paragraph on GJA5 deletions, after this there is just a summary gene, each on their own sentence. This reads rather jarring in comparison to the previous part. I would put a connecting or explanatory sentence here, to prevent it reading like a summary.

We edited this paragraph (lines 318-332) to improve readability as requested.

Reviewer 2

1. Cohort numbers are confusingly presented. For EBAV 272 probands are mentioned in the abstract, 293 families on Page 6, a cohort size of 544 (comprising both affected and unaffected samples) presented in Table 1 and 279 probands in Table 2. For BAVGWAS the text refers to 5040, but Table 2 summarizes just 3141. It is therefore unclear how many samples have contributed to the analyses. Can the authors please explain and address this so that numbers analyzed are clearly presented?

We updated lines 168-171 to summarize the EBAV and BAGWAS cohorts as follows: “The EBAV cohort included a total of 544 samples: 272 EBAV probands, 21 relatives with BAV, and 251 apparently unaffected family members (26 trios and 15 multiplex families). The BAVGWAS sample contained 5,040 genotypes with associated demographic and clinical data.” We edited Table 1 (formerly Table 2) to reflect the total number of unrelated individuals that were included in the final analysis, rather than the total numbers in each dataset. We clarified this in lines 171-172: “After exclusions due to data quality control or missing phenotypic data, 499 EBAV genotypes and 4216 BAVGWAS genoytpes were included in the final analysis.”

2. I don’t see the need for Table 1 with the information included.

We deleted Table 1.

3. Page 9, paragraph 2. Presentation of Tables 3 and S2 in relation to the text could be improved. Is the “prevalence of large and rare CNVs”, referring to the P-value for number of CNVs per individual?

We edited this paragraph (lines 181-184) as follows: “In comparisons between EBAV and WLS data, the rate of large CNVs was increased in EBAV cases compared to WLS controls, driven primarily by enrichment of large rare genomic deletions (P<0.001, Table 2 [formerly Table 3).” 

4. Table S3 could include a column listing affected genes.

We added a column listing affected genes to Table S4 as requested.

5. Where is the data supporting the page 5 statement “Seven of these genic CNVs were

enriched in EBAV cases compared to WLS controls with a genome-wide adjusted empiric P < 0.05.”

We added specific data about these enriched CNVs with empiric P-values to Tables S6 and S8.

6. Which BAV genes are in the CNVs assessed in Table 4?

We added a list of genes known to cause BAV when mutated to Table S4.

7. Page 10. “Large duplications involving the Velocardiofacial (VCFS) region in 22q11.2 and 1q21.1 microduplications were also enriched in EBAV cases (Table S4)”. The table does not present evidence for enrichment.

We added enrichment data for these rare EBAV CNVs to Table S6.

8. Which CHD genes were scrutinized? A supplementary table listing genes assessed would be helpful. I think important to show CHD associated genes where no CNV was identified.

We revised lines 162-165 in the Methods section as follows: “CNV overlap functions in PLINK were used to identify rare CNVs that intersect between datasets or involve specific BAV or CHD genes (S2 Table). The list of candidate genes included 190 CHD genes that have strong cumulative evidence to cause BAV or related congenital malformations from human or animal model data.” The specific CHD candidate genes are included in Table S2.

9. Page 10, Table 5 and table S6. I believe “subtle” should be properly defined to clarify criteria of the CNVs analysed in this section. Relationship of the two tables is unclear – for some genes (e.g., MYH11) the CNVs described in Table S6 appear to correspond to the numbers presented in Table 5, however, this is not true for others (e.g., GATA4). Please explain.

We edited lines 222-224 as follows, removing ‘subtle’ from the description: “We also scrutinized candidate genomic regions that are implicated in CHD by analyzing data from individual CNV algorithms to detect copy number alterations that may have been filtered out due to strict quality control criteria prior to enrichment studies.” The top candidate genes that are mentioned in the text and in Table 5 are marked with asterisks in Tables S6 and S8.

10. Burden testing P-values should be presented to the data in Table 5 and Table 6.

We added P-values to Table 5 and Table 6 as requested.

11. Burden testing method is not properly described.

We edited lines 159-160 as follows to describe the burden testing method: “Rare CNV functions in PLINK (v1.7) were used to perform permutation-based burden tests or gene set-based enrichment tests as previously described [28, 29].”

12. Abstract and Page 13, opening paragraph of discussion. The authors report potentially pathogenic variants for 8% of BAV cases in their EBAV cohort. It is not immediately clear to me which CNVs were considered as likely pathogenic. A table summarizing the cases where CNVs have been interpreted in this way, with a clinical description of proband/family could be included.

We edited lines 224-229 in the Results section on page 13 as follows: “We identified additional rare EBAV CNVs that intersect with CHD candidate genes CELSR1, GJA5, RAF1, LTBP1, KIF1A, MYH11, TTN, and the VCFS region in 22q11.2. We detected additional GATA4 and DSCAM CNVs in multiplex families. These CNVs were enriched in EBAV cases compared to WLS controls (Table 5, S10 Table). In total, 9% of EBAV probands had CNVs that are likely to contribute to the development of BAV (S11 Table).” Table S11 contains phenotypic data on these individuals.

13. The methods should include a section clearly describing the statistical tests performed.

We edited the appropriate section of the Methods (lines 157-160) to include descriptions of statistical methods. We edited the legends of Tables 5 and 6 to state how P values are calculated.

Reviewer 3

1. Did the authors check whether large and rare CNVs tend to occur more often in patients with BAV developing an aortic aneurysm or is the distribution equal?

In lines 204-206, we added: “EBAV probands with large rare CNVs were more likely to have concomitant congenital heart lesions and have other family members who were diagnosed with BAV (S5 Table).” Table S5 summarizes these phenotypic differences. There were no differences in the prevalence of aneurysms, which are common in EBAV cases.

2. Where the CNVs more common in families with more than 1 affected individual?

As stated in #1, EBAV probands with rare CNVs were more likely to have a relative with BAV. 

3. Interestingly, CNVs only affected the TGF-beta family in late onset BAV. How do the authors explain this finding?

There were only two rare CNVs involving TGFBR1 and TGFBR2 in the BAVGWAS dataset and they were not significantly enriched compared to controls. Deletions or duplications of TGFBR1 and TGFBR2 may cause different phenotypes than amino acid substitutions that cause Loeys-Dietz syndrome predisposing to TAD and BAV. Without additional data, it is not possible to evaluate the significance of these observations.

4. The average clinical reader may not be used to CNVs. I therefore recommend including a simple graphical abstract.

We inserted a graphical abstract (Figure 1) into the manuscript as requested.

---

## [Decision Letter · Decision Letter 1]

14 May 2024

Rare Genomic Copy Number Variants Implicate New Candidate Genes for Bicuspid Aortic Valve

PONE-D-23-37920R1

Dear Dr. Prakash,

We’re pleased to inform you that your manuscript has been judged scientifically suitable for publication and will be formally accepted for publication once it meets all outstanding technical requirements.

Kind regards,

Nejat Mahdieh

Academic Editor

PLOS ONE

Additional Editor Comments (optional):

Reviewers' comments:

Reviewer's Responses to Questions

**Comments to the Author**

1. If the authors have adequately addressed your comments raised in a previous round of review and you feel that this manuscript is now acceptable for publication, you may indicate that here to bypass the “Comments to the Author” section, enter your conflict of interest statement in the “Confidential to Editor” section, and submit your "Accept" recommendation.

Reviewer #1: All comments have been addressed

Reviewer #3: All comments have been addressed

2. Is the manuscript technically sound, and do the data support the conclusions?

Reviewer #1: Yes

Reviewer #3: Yes

3. Has the statistical analysis been performed appropriately and rigorously? 

Reviewer #1: Yes

Reviewer #3: Yes

4. Have the authors made all data underlying the findings in their manuscript fully available?

Reviewer #1: Yes

Reviewer #3: Yes

5. Is the manuscript presented in an intelligible fashion and written in standard English?

Reviewer #1: Yes

Reviewer #3: Yes

6. Review Comments to the Author

Reviewer #1: All my comments have been addressed, the manuscript has improved and is of interest. I have no further suggestions.

Reviewer #3: The reviewers comments have been fully addressed. Therefore, I am happy to recommend the manuscript for publication.

7. PLOS authors have the option to publish the peer review history of their article (what does this mean?). If published, this will include your full peer review and any attached files.

Reviewer #1: No

Reviewer #3: No

---

## [Editor Report · Acceptance letter]

28 Aug 2024

PONE-D-23-37920R1 

PLOS ONE

Dear Dr. Prakash, 

I'm pleased to inform you that your manuscript has been deemed suitable for publication in PLOS ONE. Congratulations! Your manuscript is now being handed over to our production team.

Kind regards, 

on behalf of

Dr. Nejat Mahdieh 

Academic Editor

PLOS ONE